# Multivalent Epigraph Hemagglutinin Vaccine Protects against Influenza B Virus in Mice

**DOI:** 10.3390/pathogens13020097

**Published:** 2024-01-23

**Authors:** Erika Petro-Turnquist, Brigette Corder Kampfe, Amber Gadeken, Matthew J. Pekarek, Eric A. Weaver

**Affiliations:** 1Nebraska Center for Virology, School of Biological Sciences, University of Nebraska-Lincoln, Lincoln, NE 68583, USA; 2Science Department, North Arkansas College, Harrison, AR 72601, USA; 3College of Agricultural Sciences and Natural Resources, University of Nebraska-Lincoln, Lincoln, NE 68583, USA; agadeken2@huskers.unl.edu

**Keywords:** influenza B virus, yamagata lineage, victoria lineage, epigraph, vaccine

## Abstract

Influenza B virus is a respiratory pathogen that contributes to seasonal epidemics, accounts for approximately 25% of global influenza infections, and can induce severe disease in young children. While vaccination is the most commonly used method of preventing influenza infections, current vaccines only induce strain-specific responses and have suboptimal efficacy when mismatched from circulating strains. Further, two influenza B virus lineages have been described, B/Yamagata-like and B/Victoria-like, and the limited cross-reactivity between the two lineages provides an additional barrier in developing a universal influenza B virus vaccine. Here, we report a novel multivalent vaccine using computationally designed Epigraph hemagglutinin proteins targeting both the B/Yamagata-like and B/Victoria-like lineages. When compared to the quadrivalent commercial vaccine, the Epigraph vaccine demonstrated increased breadth of neutralizing antibody and T cell responses. After lethal heterologous influenza B virus challenge, mice immunized with the Epigraph vaccine were completely protected against both weight loss and mortality. The superior cross-reactive immunity conferred by the Epigraph vaccine immunogens supports their continued investigation as a universal influenza B virus vaccine.

## 1. Introduction

Severe influenza virus infections can cause seasonal epidemics and worldwide pandemics. Humans are susceptible to two main types of influenza viruses, influenza A virus (IAV) and influenza B virus (IBV) [1]. IBV is responsible for ~25% of all yearly influenza cases [2] but can be the predominantly circulating influenza type during some seasons [3,4]. Further, epidemic IBV infections can cause a second wave of infections when IAV cases have waned [5,6]. IBV has been characterized to impose disease just as severe as IAV [7,8,9] and is often a concern for at-risk populations such as children [10,11,12,13] and the elderly [14,15]. Influenza B viruses are categorized into two antigenically distinct lineages, B/Yamagata-like and B/Victoria-like, with limited antibody cross-reactivity between the two lineages [16]. Consequently, in 2013 the CDC recommended the development of quadrivalent vaccine formulations containing representative IBV strains from both lineages [17]. 

Commercial vaccines commonly direct immunity towards the viral hemagglutinin (HA) surface glycoprotein to induce neutralizing antibody responses that block viral entry into host cells. However, continual antigenic drift of the HA protein results in viral immune escape and limited protection against antigenically mismatched strains [18,19]. Consequently, yearly vaccine efficacy against IBV ranges from 34–76% depending on the lineage predominantly in circulation [20]. To improve upon this limitation, recent research has focused on developing broadly cross-protective vaccines against both the Victoria- and Yamagata-like IBV lineages. One strategy includes peptide-based vaccines targeting the conserved cleavage site and fusion peptide of the HA_0_ polypeptide [21]. Because the cleavage site and fusion peptide of the HA protein play a critical role during the initial stages of infection, this vaccine strategy posits that eliciting antibodies towards this region will prevent infection. Indeed, following immunization with the B/HA_0_ peptide, mice were completely protected from challenge with IBV isolates derived from both the Victoria-like and Yamagata-like lineages [21]. Another strategy of developing a broadly protective vaccine against IBV is a chimeric HA protein approach. This approach uses sequential immunization with HA proteins encoding the IBV stalk and head domains derived from exotic avian IAV to drive immunity towards the more conserved stalk domain. Stalk-directed antibodies were shown to protect from lethal challenge with both Victoria-like and Yamagata-like IBV through Fc-mediated effector functions [22]. Additional efforts have refined this strategy by developing mosaic HA vaccines. In contrast to the chimeric HA approach, the mosaic vaccine only replaces the major antigenic sites of the IBV HA with exotic IAV to retain noncanonical epitopes in the head and stalk domains. These mosaic IBV HA proteins were similarly able to prevent mortality in mice and reduce morbidity through non-neutralizing antibody functions [23]. While previous research has primarily focused on the induction of cross-reactive antibody responses to provide broad protection, increasing evidence indicates that cellular-mediated immunity plays a key role in improving cross-reactive responses when antibody responses have failed [24,25,26,27]. Here, we characterize a computationally derived Epigraph HA design to induce broad humoral and cell-mediated immunity against Victoria-like and Yamagata-like IBV lineages.

The Epigraph vaccine antigen designer uses a graph-based algorithm to create vaccine antigens with maximized potential epitope coverage of a highly diverse sequence population. We have recently shown that the Epigraph HA design can induce significantly improved cross-reactive responses against swine H3 [28] and human H3 influenza A virus [29] than those induced by commercial vaccines and a wildtype comparator vaccine. Here, we build on this previous success to evaluate the protective efficacy of Epigraph immunogens designed against the HA protein of IBV (IBV-Epi). We characterized the Epigraph immunogens, evaluated the cross-reactivity of the vaccine antigens to induce antibody and T cell responses against both the Victoria-like and Yamagata-like IBV lineages, and assessed protection against challenge in mice. When compared to a commercial vaccine, Fluzone, IBV-Epi significantly outperformed this current standard-of-care vaccine. These data support the use of Epigraph immunogens to provide broad protection against IBV.

## 2. Materials and Methods

### 2.1. Ethics Statement

Female BALB/c mice (aged 6–10 weeks) were obtained from Jackson Laboratory. Mice were housed in the Life Sciences Annex building on the University of Nebraska—Lincoln (UNL) campus under the Association for Assessment and Accreditation of Laboratory Animal Care International (AAALAC) guidelines. The protocols were approved by the UNL Institutional Animal Care and Use Committee (IACUC) (Project ID 2158: Influenza Vaccine Development). All animal experiments were carried out according to the provisions of the Animal Welfare Act, PHS Animal Welfare Policy, the principles of the NIH Guide for the Care and Use of Laboratory Animals, and the policies and procedures of UNL (IBC protocol: 619). Immunizations, bleeds, and experimental infections were performed under isoflurane or ketamine and xylazine-induced anesthesia.

### 2.2. Viruses

Influenza strains (B/Malaysia/2506/04 NR-12280; B/Florida/04/06 NR-9696) were provided by Biodefense and Emerging Infections Research Resources Repository NIAID. Influenza strains (B/Nevada/03/11 FR-1028; B/Texas/06/11 FR-1062; B/Phuket/3073/13 FR-1364; B/Pennsylvania/07/07 FR-16; B/Victoria/304/06 FR-20; B/Texas/26/08 FR-360; B/Colorado/06/17 FR-1588) were provided by the International Reagent Resource. Viral stocks were inoculated into specific pathogen free (SPF) embryonated chicken eggs after 11 days of incubation and harvested three days after inoculation. The virus was collected by harvesting allantoic fluid and centrifuged at 200× *g* for 10 min to separate egg protein from grown virus stocks. Aliquots of supernatant were stored at −80 °C. Viruses were quantified based on HAU and TCID_50_. The B/Phuket/3073/2013, B/Florida/04/2006, B/Washington/2/2019, and B/Malaysia/2506/2004 influenza B viruses were mouse adapted through serial lung passaging in mice 10 times. The adapted viruses were collected from the final passage of mouse lungs, grown in SPF embryonated chicken eggs, and prepared and quantified as specified above. The HA gene of B/Phuket/3073/2013 and B/Florida/04/2006 were sequenced as previously described [30] to ensure no mutations had occurred during serial lung passaging. 

### 2.3. Immunogen Gene Design 

The human influenza B Epigraph HA immunogens were designed with Epigraph Vaccine Designer (Los Alamos National Laboratories Database). All unique human influenza B HA protein sequences which included the keywords “Victoria” (743 sequences) or “Yamagata” (579 sequences) were downloaded from the Influenza Research Database. The resulting 743 unique Victoria HA sequences were aligned in ClustalX2.1 and submitted to the Epigraph program in fasta format with the following parameters: Vaccine Pool: 3, Epitope Length: 9, Rare Threshold: 2. Likewise, the 579 unique Yamagata HA sequences were aligned and submitted to the Epigraph program with the same parameters. 

### 2.4. Phylogenetic and Sequence Analysis 

All 1322 unique influenza B HA protein sequences from 1940 to 2020 were aligned with Geneious 11.0.5. The output alignment file was used in Geneious to design a neighbor joining tree using the Jukes–Cantor model. All immunogens and relevant HA strains were labelled on the phylogenetic tree. The relevant HA strains and immunogens were aligned with Geneious and used to make a simplified neighbor joining tree. The percent similarity using Blosum62 cost matrix with threshold 1 and percent identity between the HA protein sequences was calculated using Geneious 11.0.5. 77 The predicted epitope coverage for each vaccine was calculated using the EpiCover program (Epitope Coverage Assessment Tool; Los Alamos National Laboratories Database). The Epigraph HA immunogens were analyzed as a hexavalent collection of vaccine sequences for potential T and B cell epitope (PTBE) coverage with the following parameters: normal epitope length = 9; maximum amino acid mismatches to score = range 0–2; minimum number of occurrences of a potential epitope in viral protein set to consider for coverage = 3; and decimal places when reporting coverage = 4. Epitope coverage was predicted for all 1322 unique influenza B HA, the 579 Yamagata-like HA sequences, and the 740 Victoria-like HA protein sequences. 

### 2.5. Recombinant Adenovirus Type 5 Plasmid Construction and Confirmation 

Each Epigraph immunogen was cloned into a recombinant adenoviral expression vector as previously described [31]. Briefly, the HA genes were optimized for human gene expression then cloned into recombinant replication deficient human Adenovirus type 5 (Ad5) using the AdEasy Adenoviral Vector System (Agilent, Santa Clara, CA, USA). Successful recombinants were confirmed through restriction enzyme digest and sequencing. Recombinant Ad5-HA clones were transfected, grown, and amplified by sequentially passaging viral stocks in E1 complementing 293 cells. High viral titer stocks were obtained after a final amplification step in a Corning 10-cell stack flask. High titer viral stocks were purified with two sequential CsCl gradients, then desalted with Econo-Pac 10DG Desalting Columns (Bio-Rad). Viral stocks were stored at −80 °C in Ad-tris buffer with 10% glycerol. The virus particle quantity was measured with a NanoDrop Lite spectrophotometer at OD260. Infectious units were quantified by Adeno-X Rapid Titer Kit according to manufacturer’s instructions (Takara Bio Company, San Jose, CA USA) (Appendix A).

### 2.6. Epigraph Immunogen Rosette Analysis 

Rosette analysis was carried out as described previously [32] with some modification. A549 cells (2.5 × 10^5^ cells/mL) were seeded into a 6-well plate and incubated at 37 °C for 2 h to allow cell adherence to the well. The media was removed, and each adenovirus constructed was added to the respective well at 500 vp/cell. Plates were rocked at 37 °C for 1 h, then DMEM containing 5% FBS was added to each well. Twenty-four hours after infection, the wells were washed with DPBS twice and 1 mL of 1 × 10^7^ chicken red blood cells/mL were added to each well. Plates were incubated at 37 °C for 40 min, then washed once with DPBS. Rosettes were visualized on an EVOS FL Cell Imaging System (Thermo Fisher, Waltham, MA USA).

### 2.7. Western Blotting

Expression of HA proteins in Ad5 was determined by Western blot. Confluent 293 cells were infected with 500 vp/cell of recombinant adenovirus individually expressing each Epigraph HA and incubated for 24 h prior to harvest. To obtain HA monomer proteins, cells were treated with Laemmli buffer with 2-mercaptoethanol and heated at 100 °C for 10 min. To obtain HA proteins in native trimer form, cells were resuspended in Laemmli buffer then passed through a QiaShredder. Samples were loaded onto a 12.5% (denatured HA) or 7.5% (native HA) SDS-PAGE gel then transfered onto a nitrocellulose membrane. The membrane was blocked in tris-buffered saline and Tween 20 (TBST) with 5% non-fat milk for 30 min. Membranes were then incubated overnight at 4 °C in polyclonal goat anti-HA B/Hong Kong/8/1973 antibody (NR-3165; BEI resources) or mouse anti-GAPDH-HRP (sc-47724; Santa Cruz Biotechnology, Inc., Dallas, TX, USA) at 1:1000 in TBST with 1% milk. The membrane was washed three times in TBST then incubated with donkey anti-goat IgG-HRP conjugated antibody (Millipore Sigma, Burlington, MA, USA) at 1:10,000 in TBST with 1% milk for one hour at room temperature. After three washes with TBST, the membrane was developed with Super Signal West Pico Chemiluminescent Substrate (Thermo Scientific, Waltham, MA, USA) and imaged on a ChemiDoc (BioRad, Hercules, CA, USA). 

### 2.8. Tissues for Humoral and Cellular Assays 

Female BALB/c mice were immunized with 10^10^ vp of IBV-Epi, 600 ng of Fluzone (approximately 30× an equivalent human dose based on body mass), or control phosphate buffered saline (DPBS). The hexavalent Epigraph vaccine was a cocktail of 1.67 × 10^9^ vp for each respective HA component (B/Victoria/Epigraph 1: 3.01 × 10^7^ infectious units; B/Victoria/Epigraph 2: 8.68 × 10^6^ infectious units; B/Victoria/Epigraph 3: 7.92 × 10^6^ infectious units; B/Yamagata/Epigraph 1: 6.09 × 10^6^ infectious units; B/Yamagata/Epigraph 2: 4.23 × 10^6^ infectious units; B/Yamagata/Epigraph 3: 7.69 × 10^6^ infectious units). Mice were immunized twice at three-week intervals. Two weeks after, the boost immunization serum and spleens were collected. Blood was centrifuged at 6000× *g* for two minutes in Becton Dickinson microtainer blood collection tubes and collected sera samples were used for hemagglutination inhibition (HI) assay and virus neutralization assay. Single-cell splenocyte samples were collected by passing spleens through a 40 µm nylon cell strainer then lysing red blood cells with ACK lysis buffer (150 mM NH4Cl, 10 mM KHCO_3_, 0.1 mM Na_2_EDTA). Single-cell suspensions of splenocytes were used in a downstream enzyme linked immunospot assay.

### 2.9. Hemagglutination Inhibition Titers

Serum samples were analyzed by hemagglutination inhibition (HI) assay. Serum samples were treated with receptor destroying enzyme (Denka Seiken) at a 1:3 ratio at 37 °C. After overnight incubation, serum samples were heat-inactivated at 56 °C for 30 min then serially diluted two-fold in a 96-well V-bottom plate. Then, 8 hemagglutinating units (HAU) of respective virus were added to all wells and incubated at room temperature for 1 h, then 0.5% chicken red blood cells were added to analyze hemagglutination patterns. HI titers were performed with virus stocks provided by the International Reagent Resource (IRR) or the Biodefense and Emerging Infections Research Resources Repository NIAID (BEI Resources).

### 2.10. Microneutralization Titers

Sera from vaccinated animals was measured for virus neutralization. Briefly, sera samples were diluted 2-fold with DPBS in sterile 96-well U-bottom plates before the addition of 50 TCID_50_ virus to each well. Approximately 100 µL of 2 × 10^5^ MDCK cells/mL were added to each well after 1 h of incubation at 37 °C. After overnight incubation at 37 °C and 5% CO_2_, cells were washed once with 200 µL DPBS/well before replacing the media with DMEM containing 2 µg/mL TPCK trypsin. Plates were incubated for 72 h at 37 °C and 5% CO_2_, and 50 µL of 0.5% chicken red blood cells were added to each well and read for hemagglutination after 1 h. 

### 2.11. ELISpot Assay 

The total T cell response to immunization was measure by IFN-γ ELISpot assay using peptide arrays (B/Nanchang/12/98 NR-2605; B/Florida/04/06 NR-18972; B/Brisbane/60/08 NR-19247; B/Malaysia/2506/04 NR-18967) provided by Biodefense and Emerging Infections Research Resources Repository NIAID. All peptide arrays spanned the entire HA gene and consisted of 13 to 18-mers with 9 or 12 amino acid overlaps. Polyvinylidene difluoride-backed 96-well plates were coated at 4 °C overnight with 5 µg/mL of anti-mouse IFN-γ AN18 monoclonal antibody (Mabtech, Cincinnati, OH, USA). Plates were then washed 4 times with DPBS and blocked in RPMI containing 5% FBS at 37 °C for one hour. Single-cell suspensions of splenocytes (2 × 10^5^ cells/well) and peptides (5 µg/mL), concanavalin A (ConA; 5 µg/mL), or RPMI were added to the wells then splenocytes were stimulated overnight at 37 °C with 5% CO_2_. After 16–18 h of overnight incubation, wells were washed 6 times with DPBS and 100 µL of biotinylated anti-mouse IFN-γ R4-6A2 (Mabtech) diluted to 1:1000 in DPBS with 1.0% FBS was added to all wells. After incubation at room temperature for 1 h, plates were washed 6 times with DPBS, and streptavidin-alkaline phosphatase conjugate diluted 1:1000 in DPBS with 1.0% FBS was added to all wells and incubated at room temperature for 45 min. Plates were washed an additional 6 times, then developed with BCIP/NBT (Plus) alkaline phosphatase substrate (Thermo Fisher). Once spots were observed in the ConA positive control wells, the developing reaction was stopped by washing the wells several times with ddH_2_O. Spots were counted on an automated ELISpot plate reader (AID iSpot Reader Spectrum; Autoimmun Diagnostika GmbH) and total T cell responses are represented as spot-forming units (SFU) per 10^6^ splenocytes.

### 2.12. Vaccination and Lethal IBV Challenges in Mice 

Groups of female BALB/c mice (*n* = 5) were vaccinated with 10^10^ vp of the IBV-Epi (1.67 × 10^9^ vp per Epigraph HA), 600 ng of Fluzone, or with DPBS as a negative control vaccination. All mice were immunized by intramuscular injection with 25 µL of vaccine per hind leg. Mice were primed, boosted three weeks later, then challenged intranasally with 100MLD_50_ of mouse-adapted B/Phuket/3073, B/Florida/04/2006, B/Washington/2/2019, or B/Malaysia/2506/2004. The mice were monitored for weight loss for two weeks and mice that lost ≥25% of their initial weight were humanely sacrificed by CO_2_ asphyxiation and cervical dislocation. 

### 2.13. Statistical Analysis

Data was analyzed using GraphPad Prism 9.1.0 software. Data are expressed as the mean with standard error (SEM). HI, microneutralization, ELISpot data, and area under the curve (AUC) of weight loss plots were analyzed by one-way ANOVA with Tukey’s multiple comparison follow-up test. A *p*-value < 0.05 was considered statistically significant (* *p* < 0.05; ** *p* < 0.01; *** *p* < 0.001; **** *p* < 0.0001).

## 3. Results

### 3.1. Design and Characterization of the IBV Epigraph Immunogens

The Epigraph vaccine designer is tool that uses a graph-based algorithm to create a cocktail of synthetic immunogens with maximized potential epitope coverage from a population of viral sequences [33,34]. When a population of viral sequences with high sequence diversity is used, the Epigraph vaccine designer calculates the frequency of each 9-mer epitope in the target population and uses a graph-based algorithm to link contiguous epitopes across the HA protein. The synthetic, full-length HA proteins produced contain the most common potential 9-mer epitopes in the target virus population. The IBV Epigraph immunogens were developed to target both the Yamagata-like and Victoria-like IBV lineages. All 743 unique IBV HA protein sequences that included the keyword “Victoria” were used to design 3 Epigraph HA immunogens containing the highest number of 9-mer potential epitopes in the Victoria-like IBV lineage. Likewise, the 579 unique IBV HA protein sequences that included the keywork “Yamagata” were used to design 3 Epigraph HA immunogens against the Yamagata-like IBV lineage. We generated a neighbor-joining phylogenetic tree of all unique IBV HA protein sequences to assess the phylogenetic relationship between the Epigraph HA proteins (purple), strains included in the commonly used seasonal commercial vaccine, Fluzone (blue), and IBV sequences (Figure 1A). The Yamagata Epigraph 1 (B/Yamagata/Epigraph/1) localized in the center of the main Yamagata-like cluster, while the Yamagata Epigraph 2 (B/Yamagata/Epigraph/2) localized in an adjacent Yamagata-like cluster. The third Yamagata Epigraph (B/Yamagata/Epigraph/3) localized towards the Yamagata strains but did not localize within observed sequence clusters. As expected, the first Victoria Epigraph (B/Victoria/Epigraph/1) was found in the center of the Victoria-like IBV strains. In contrast, the second Victoria Epigraph (B/Victoria/Epigraph/2) localized towards the Yamagata sequences. This could be due to sequence similarity between the Victoria strains and the adjacent Yamagata cluster or may be an artifact of the using keyword search rather than phylogeny when defining inclusion criteria for the vaccine design. Similar to the B/Yamagata/Epigraph/3 HA, the third Victoria Epigraph (B/Victoria/Epigraph/3) sequence was divergent from IBV sequences but localized within the respective Victoria cluster. All six Epigraph genes were individually cloned into a replication-defective adenovirus type 5 viral vector and HA protein expression was assessed by Western blot analysis (Figure 1B). Epigraph HA protein expression was analyzed under both native and denatured conditions and exhibited similar levels of expression among each Epigraph. Given that the Epigraphs are synthetic in design, we next wanted to assess the structure and folding of the encoded protein. All six Epigraph proteins were submitted to the SWISS-MODEL web server to predict the structure (Figure 1C). This server uses homology-based structural modeling between the input sequence and a template with known structure to predict folding [35,36]. Structural analysis revealed that the synthetic Epigraph proteins maintained appropriate folding of the 140 loop, 160 loop, 190 helix, and the 240 loop composing the receptor binding site of the HA protein [37]. Other structural features, such as the fusion protein and the transmembrane domain, were also retained in the synthetic design. Finally, we assessed the ability of the Epigraphs to exhibit functional hemagglutination capacity. To this end, the Epigraph-expressing adenoviruses were used to infect A549 cells and assessed for rosette formation of sialic-acid expressing red blood cells (Figure 1D, Appendix A). Representative images of both Victoria and Yamagata Epigraph 1 are shown, and the HA-expressing A549 cells exhibited robust binding to red blood cells while the uninfected controls did not produce rosettes indication that the epigraph constructs maintained normal hemagglutinin function. 

### 3.2. Bioinformatic Predictions of IBV Epigraph Immunogen Cross-Reactivity

To assess the relationship of strains and vaccines used throughout this study, the HA protein sequences were compared based on sequence similarity and percent identity (Figure 2A). A heatmap and neighbor-joining tree shows the relationship of these HA sequences and a clear distinction between the Yamagata-like and Victoria-like IBV strains. As expected, all three Yamagata Epigraph HA proteins, and the first and third Victoria Epigraph HA proteins were genetically similar to their respective IBV lineage. In contrast, the second Victoria Epigraph HA shared more genetic similarity to the Yamagata-lineage strains. To analyze the percent epitope coverage between the strains included in the study vaccines and all unique IBV strains, an Epitope Coverage Plot (EpiCover) was created (Figure 2B). This tool determines the fraction of *k*-mers shared between target vaccine strains and a population of diverse sequences to identify shared, rare, or absent *k*-mer epitopes [38]. Given that similar epitopes have the potential to cross-react, both the exact-matched (Off-by-0) and near-matched scores (Off-by-1, Off-by-2, and Off-by > 2) are computed for each target vaccine antigen. Analysis of the two vaccine strains included in the 2018–2019 Fluzone formulation, B/Phuket/3073/2013 and B/Colorado/06/2017, revealed that 91% of epitopes were exact-matched and 9% of epitopes were near-matched to all unique Yamagata-lineage and Victoria-lineage IBV strains. In contrast, the IBV-Epi vaccine strains contained exact-matched epitopes with 96.86% of all unique Yamagata-lineage and Victoria-lineage IBV strains and 3.14% of near-matched epitopes. The higher percentage of exact-matched epitopes compared to near-matched epitopes suggests that the Epigraph may induce broader cross-reactive immunity compared to a commonly used standard-of-care commercial vaccine. 

### 3.3. IBV Epigraph Vaccine Elicits Broad Humoral Immune Responses

To evaluate the immunogenicity of IBV Epigraph (IBV-Epi), mice were intramuscularly immunized twice with IBV-Epi, a seasonal commercial comparator vaccine (Fluzone; 2018–2019 formulation), or DPBS sham vaccine. Serum samples were collected after the boost immunization and assessed for antibody responses against a panel of divergent Yamagata-like and Victoria-like IBV strains. Vaccination with IBV-Epi elicited protective HI antibody responses (≥40; log_2_: 5.32) [39,40] against 4 out of the 9 (44%) representative IBV strains, while Fluzone only elicited protective HI antibody responses against 2 out of the 9 (22%) representative IBV strains analyzed (Figure 3A). Interestingly, we observed that both the magnitude and breadth of HI antibody responses against representative Victoria-lineage isolates were limited compared to the HI antibody responses against the analyzed Yamagata-lineage isolates. This observation may be due to the higher number of vaccine antigens that localize in the Yamagata-lineage (Figure 1A). While the HI assay is considered the gold standard immunological measurement for serum antibody responses, a limitation of the HI assay is constriction to only identifying antibodies that block the receptor binding site of the HA molecule on the surface of IBV. Consequently, we analyzed serum antibody responses by virus neutralization assay. Virus neutralization assays quantify any functional antibody responses that directly block infection of a eukaryotic cell to prevent cytopathic effects in vitro and exhibits higher sensitivity than a standard HI assay [41,42]. As expected, the overall magnitude of neutralizing antibody responses was higher than those observed by HI assay. We observed that mice immunized with IBV-Epi induced robustly neutralizing antibody responses against all representative Yamagata-like and Victoria-like IBV strains tested (Figure 3B). Notably, IBV-Epi was able to elicit significantly higher neutralizing antibody responses compared to Fluzone against B/TX/06/11, B/PA/7/07, B/WA/2/19, and B/NV/03/11, indicating the potential for improved protection against these strains. Overall, these data indicate that immunization with IBV-Epi produces significant cross-reactive antibody responses that were functionally capable to inhibit and neutralize diverse strains of IBV.

### 3.4. IBV Epigraph Vaccine Provides Broadly Cross-Reactive Cellular-Mediated Immune Responses

Cross-reactive T cell responses targeting influenza play a key role in the clearance of viral infected cells [24,25,26,27]. Therefore, we next assessed the induction of T cell responses to two Victoria-lineage and two Yamagata-lineage isolates after boost immunization by IFN-γ ELISpot assay. Splenocytes were collected two weeks after boost immunization and stimulated ex vivo with overlapping peptide arrays spanning the entire HA protein of B/Malaysia/04, B/Brisbane/08, B/Nanchang/98, and B/Florida/06 (Figure 4). Consistent with previous findings, Fluzone induced a modest T cell response against all IBV strains analyzed (Malaysia/04 mean: 180 SFU/10^6^ cells; Brisbane/08 mean: 340 SFU/10^6^ cells; Nanchang/98 mean: 172 SFU/10^6^ cells; Florida/06 mean: 231 SFU/10^6^ cells) that were not significantly higher than negative control immunized mice. In contrast, immunization with IBV-Epi induced robust cross reactive IFN-γ secreting total T cell responses against Malaysia/04 (mean: 2257 SFU/10^6^ cells), Brisbane/08 (mean: 3362 SFU/10^6^ cells), Nanchang/98 (mean: 2309 SFU/10^6^ cells), and Florida/06 (mean: 2464 SFU/10^6^ cells) that were significantly higher than mice immunized with DPBS or Fluzone. Overall, these data indicate that vaccination with IBV-Epi induces high levels of cross-reactive IFN-γ secreting T cells against representative Yamagata-lineage and Victoria-lineage IBV strains.

### 3.5. IBV Epigraph Completely Protects against Lethal IBV Challenge

We finally sought to evaluate the ability of the IBV Epigraph to protect against lethal challenge with two representative Victoria-like and two representative Yamagata-like IBV strains. Mice were immunized twice with IBV-Epi, Fluzone, or DPBS then challenged with 100 times the median lethal dose 50% (100MLD50) of divergent Victoria-lineage IBV (B/WA/2/19 or B/Malaysia/2506/04) or Yamagata-lineage IBV isolates (B/Phu/3073/13 or B/FL/04/06) and monitored for weight loss and mortality. Mice vaccinated with IBV-Epi were completely protected from weight loss and exhibited 100% survival against all four challenge strains, regardless of lineage. In contrast, mice immunized with Fluzone showed substantial weight loss and 20% of mice succumbed to infection with B/WA/2/19 (Figure 5A). Immunization with Fluzone further failed to protect against challenge with B/Malaysia/2506/04, as mice in this vaccine group demonstrated an average of 15–20% weight loss after infection. The AUC of weight loss of Fluzone-immunized mice was not significantly different than DPBS sham immunized mice, and 60% of the mice succumbed to infection during the course of infection (Figure 5B). Mice immunized with Fluzone and subsequently challenge with representative Yamagata-lineage IBV strains revealed improved protection compared to Victoria-lineage IBV challenge data. Interestingly, mice immunized with Fluzone then challenged with the matched strain, B/Phu/3073/2013, still exhibited significant weight loss compared to IBV-Epi (Figure 5C). Challenge with a more divergent Yamagata-lineage strain, B/FL/04/06, showed that Fluzone-immunized mice again exhibited more severe weight loss compared to IBV-Epi however no mortality was observed (Figure 5D). Weights of individual mice are available in Figure 2. Overall, these data indicate that IBV-Epi induces robust protection against four representative Victoria-lineage and Yamagata-lineage IBV strains that outperforms a commonly used seasonal vaccine, Fluzone.

## 4. Discussion

Influenza virus is a significant global pathogen that affects millions of individuals annually. Though IBV infections are less common than IAV infections, IBV infection can impose similarly severe disease outcomes, particularly in children and the elderly. Given that humans are the only known reservoir of IBV [43], a rigorous vaccination program using a broadly protective IBV vaccine may help in eradicating IBV. Consequently, efforts to develop a universal IBV vaccine have increased. Here, we report a vaccine strategy that utilizes a computational algorithm to maximize potential epitopes incorporated into the immunogen design and provide broad protection against antigenically diverse IBV. Our group has previously demonstrated the utility of Epigraph HA immunogens against swine H3 [28] and human H3 influenza A virus [29] and here we show similar improved cross-reactive efficacy when applied to IBV. Mice immunized with IBV-Epi developed highly cross-reactive HI and neutralizing antibody responses. Additionally, these mice developed significant T-cell responses against a panel of divergent IBV strains and were completely protected from morbidity and mortality after stringent lethal challenges with both Victoria-like and Yamagata-like IBV.

By design, the Epigraph algorithm maximizes potential epitopes incorporated into the immunogen design while retaining the conserved motifs that facilitate appropriate protein folding [34]. This protein folding plays a crucial role in the presentation of confirmational epitopes and antibody recognition of influenza [44]. When analyzing the structural characteristics of the Epigraph HA proteins we observed that they retained the highly conserved structural domains, and highly specific binding-site looks that are essential for receptor binding and viral entry. The Epigraph HA proteins also maintained the ability to trimerize and exhibited sialic acid-binding properties in vitro (Figure 1). This indicates that, though the Epigraphs are synthetic in design, they still retain the main structural and functional capabilities necessary for immune system recognition and broad cross-protection against IBV. The Epigraph strategy allows for synchronous inclusion of T and B cell epitopes despite the limited sequence space of the IBV HA protein. In addition to our modeling analysis, our immune correlate data also supports structural retention of our IBV-Epi immunogens. The Epigraph vaccine induced strong cross-reactive neutralizing antibody responses against an array of Yamagata-like and Victoria-like IBV strains isolated between 2004 to 2019. There are likely key conformational epitopes involved in the neutralization of IBV HA similar to identified epitopes in IAVs [45,46], and the protein structure would need to be retained for these epitopes to be targeted after vaccination. 

When comparing the neutralization capacity of IBV-Epi with the standard-of-care vaccine, Fluzone, we observed greater breadth and magnitude of antibody responses (Figure 3). Interestingly, we observed more robust HI antibody responses against the panel of Yamagata-like lineage IBV strains. This could be due to the B/Victoria/Epigraph/2 immunogen localizing in the Yamagata-like lineage and driving immune responses to this lineage. The localization of B/Victoria/Epigraph/2 in the Yamagata-like lineage could be explained by the inclusion of ancestral IBV sequences in the separate lineage-specific Epigraph immunogen datasets, sequence similarity between the Victoria strains and the adjacent Yamagata cluster, or may be an artifact of the method used to identify viral lineage for vaccine design. Further, an important note for this study is that direct comparison of our hexavalent Epigraph vaccine and quadrivalent Fluzone is not entirely equivalent due to the differences in valency. We have previously interrogated the contribution of individual Epigraph immunogens targeting human H3 IAV, and observed that the first and second Epigraph HA proteins induced the strongest cross-reactive antibody responses, and the third Epigraph HA protein elicited strong cross-reactive T cell responses [29]. These data indicate that each Epigraph HA protein is uniquely contributing to immunity and a trivalent cocktail is likely necessary to provide broadly cross-reactive responses. In light of these findings, additional studies are currently in progress to determine the participation of individual immunogens in protection and identify the compounding effects of using a multivalent vaccine strategy against IBV. 

In addition to strong cross-reactive neutralizing antibody responses, mice vaccinated with IBV-Epi showed robust T cell-mediated immunity against four divergent IBV strains from both the Victoria-like and Yamagata-like lineages while the inactivated split virus comparator vaccine, Fluzone, demonstrated modest induction of T cell-mediated immunity. The variation of these responses may be due to differences in the vaccine platform. Viral vectors are well known for inducing potent adaptive immunity through intracellular antigen expression and activation of MHC pathways through direct and cross-presentation of the delivered immunogen [47]. In contrast, inactivated virus vaccine platforms do not infect cells to facilitate MHC presentation and priming of CD8^+^ T cells. This often leads to limited T cell-mediated immunity against influenza after vaccination [48]. Our previous research has assessed T cell activation of the adenoviral vectored Epigraph HA proteins targeting swine H3 IAV compared to an adenoviral vectored vaccine encoding WT swine H3 IAV. While the adenoviral vectored WT HA showed significant T cell activation to the matched strain, immunization with the Epigraph HA proteins demonstrated an increased breath and magnitude of T cell responses against both swine H3 and human H3 IAV [28]. This suggests that the adenoviral vector is contributing to T cell activation, but the breadth of T cell responses against divergent strains is likely due to the Epigraph design. Additional studies directly comparing the Epigraph HA proteins to WT IBV HA proteins are necessary to determine if this same effect is observed with IBV. 

A gold standard of vaccine efficacy in the mouse model is protection against weight loss and death after a lethal viral challenge. In this study we assessed protection against two Victoria-like and two Yamagata-like IBV strains. Mice immunized with IBV-Epi showed complete protection against infection, regardless of the strain used for challenge. In contrast, mice immunized with Fluzone demonstrated significant weight loss and mortality after Victoria-like lineage challenges and partial protection against Yamagata-like lineage challenges compared to the unimmunized control mice. Interestingly, we observed that the IBV-Epi vaccine induced more robust protection against weight loss after challenge with 100MLD_50_ of the antigen-matched strain in Fluzone, B/Phuket/3073/2013 (Figure 5). This effect could be due to the stringency of the lethal challenge, where antibodies mounted against B/Phuket/3073/2013 in the Fluzone group were not high enough to neutralize and inhibit viral infection. Further, this may be an effect of the robust cross-reactive T cell responses observed in the IBV-Epi group. While we did not directly analyze T cell responses to B/Phuket/3073/2013 due to limitations in reagent availability, mounting evidence indicates that T cell responses targeting conserved viral proteins can induce protection against inter- and intra-lineage challenge [49]. Indeed, recent research has shown that T cell responses play a significant role in clearing viral infected cells when antibodies fail [24,25,26,27]. Future research using passive or adoptive transfer after immunization may help uncover the precise mechanisms of the improved protection observed.

An important development for IBV vaccine development is the possible extinction of the Yamagata-like IBV lineage [50]. The Yamagata-like and Victoria-like IBV lineages have been described to alternate in global dominance over the past several years, but currently no Yamagata-like IBV strains have been identified in circulation since March 2020. Consequently, in September 2023, the World Health Organization vaccine composition advisory committee no longer recommended including Yamagata-like IBV antigens into seasonal vaccines for the southern hemisphere [51]. The disappearance of the Yamagata-like lineage may be due to a combination of social distancing efforts, travel restrictions, and compliance with barrier measures during the COVID-19 pandemic [52]. However, it remains possible that the Yamagata-like lineage may still be in circulation at low, undetectable levels and infections may be present in isolated or rural populations. Additionally, Yamagata-like IBV strains could be present within the “undetermined” subtype of worldwide IBV infections [53]. Future global surveillance will help delineate whether this lineage has truly gone extinct or has simply entered a state of “dormancy”, as has been previously described in the 1990s with the Victoria-like lineage [54]. In the event that the Yamagata-like lineage reemerges in the future, the impact of removing this lineage from vaccine formulations could be significant, especially in children with no preexisting immunity. Therefore, development of a universal IBV targeting both lineages should be continued in the event of reemergence.

Here, we have demonstrated the efficacy of Epigraph HA vaccine immunogens to induce broadly cross-reactive immunity to Victoria-like and Yamagata-like IBV lineages. We observed robust, neutralizing antibody responses, cross-reactive T cell responses, and complete protection against four divergent challenges. We compared the Epigraph vaccine to a quadrivalent commercial vaccine, Fluzone, and observed significantly better immunogenicity and protective efficacy than the standard-of-care vaccine. Our previous work has demonstrated the utility of Epigraph HA vaccines against influenza A virus, and here we report the first published efficacy study using Epigraph IBV HA vaccine immunogens. These data support the continued development Epigraph immunogens as a broadly reactive universal influenza vaccine. 

## Figures and Tables

**Figure 1 pathogens-13-00097-f001:**
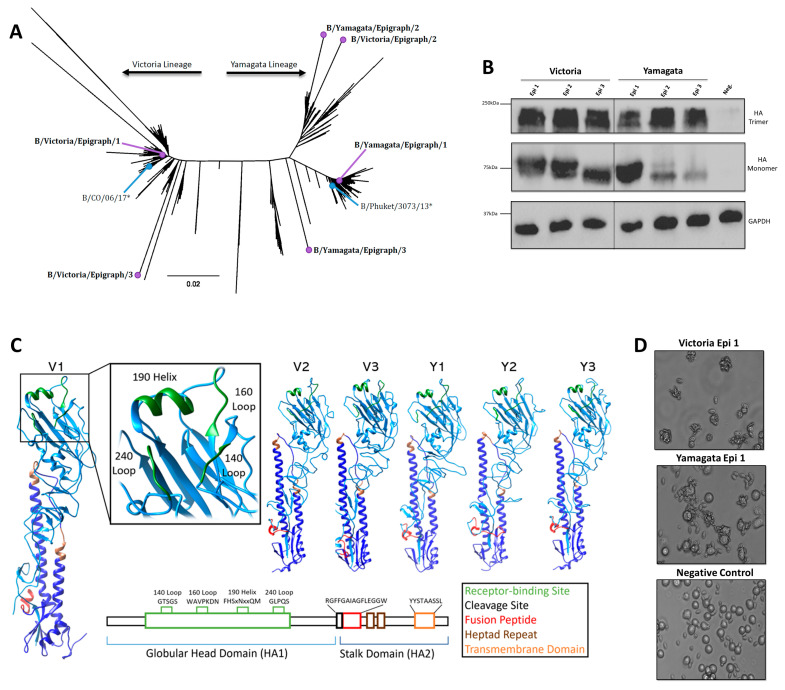
**Characterization of IBV Epigraph Immunogens.** (**A**) All unique human IBV HA protein sequences were used to make a neighbor-joining tree to visualize the phylogenetic relationship between the IBV Epigraph immunogens (purple), the strains included in the 2018–2019 commercial vaccine, Fluzone (blue), and wildtype Victoria-like and Yamagata-like IBV strains. Asterix indicates vaccine strain. (**B**) Protein expression of the HA-expressing recombinant human Ad5 vectors. Cells were infected with 500 vp per cell of HA-expressing Ad5 and protein expression was detected by western blot under native (top) and denatured (middle) conditions. GAPDH served as a cellular loading control. (**C**) Each Victoria-lineage and Yamagata-Lineage Epigraph were submitted to SWISS-MODEL and visualized with Chimera to predict protein structure of the synthetic proteins. The receptor binding site is shown in green, fusion peptide is shown in red, heptad repeat is shown in brown, and transmembrane domain is colored orange. (**D**) The synthetic Epigraph immunogens were analyzed for their ability to form trimers and bind sialic acid proteins by rosette assay. A549 cells were infected with 500 vp per cell of HA-expressing Ad5 then analyzed for binding to sialic acid-expressing red blood cells. Representative Victoria Epigraph 1 (top) and Yamagata Epigraph 1 (middle) are shown. Uninfected A549 cells served as a negative control to exhibit no rosette formation after incubation with red blood cells.

**Figure 2 pathogens-13-00097-f002:**
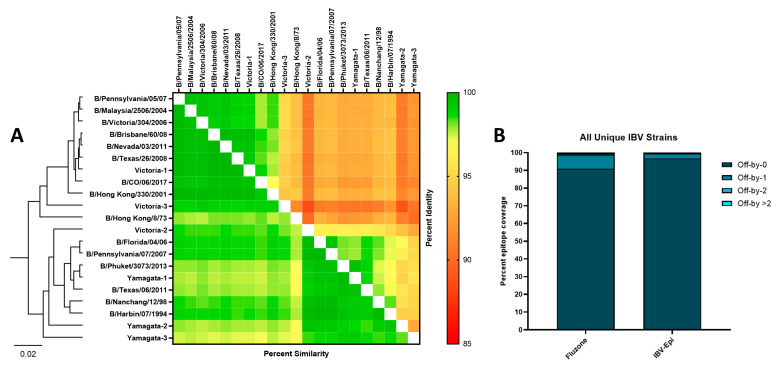
**Sequence Analysis and Epitope Coverage of Epigraph Immunogens**. (**A**) HA sequences use in the study were analyzed based on phylogeny (left), percent similarity (lower left), and percent identity (upper right). (**B**) Percent epitope coverage of strains included in Fluzone and the IBV-Epi were analyzed using the Epitope Coverage Assessment Tool (EpiCover) against all unique IBV strains. The percentage of exact-matched epitopes are indicated on each bar.

**Figure 3 pathogens-13-00097-f003:**
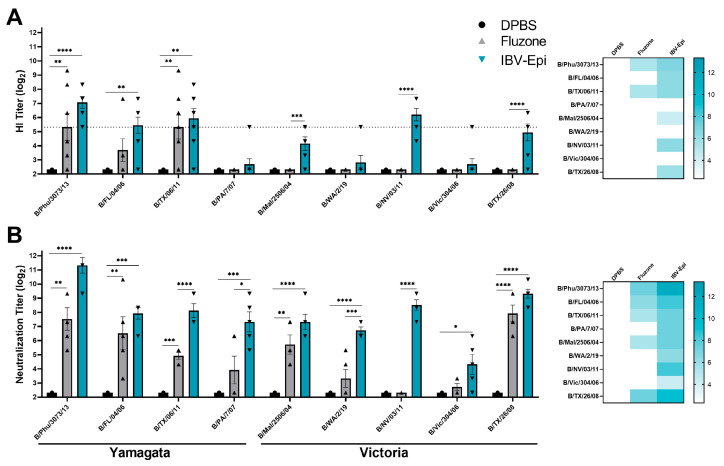
**Antibody response after vaccination.** Mice were immunized twice with IBV-Epi, Fluzone, or DPBS as a sham vaccination then serum was collected for antibody analysis. (**A**) Hemagglutination inhibition (HI) analysis against a panel of representative Yamagata-like and Victoria-like IBV strains. Dotted line: HI titer ≥ 40 (log_2_ 5.32) (**B**) Virus neutralization analysis against a panel of representative Yamagata-like and Victoria-like IBV strains. A heat map of both the HI titers and neutralization titers was constructed to further visualize the cross-reactive antibody responses elicited by each vaccine. (*n* = 5; statistical analysis was determined using a one-way ANOVA with Tukey’s multiple comparison follow up test; * *p* < 0.05, ** *p* < 0.01, *** *p* < 0.001, **** *p* < 0.0001). Data are presented as the mean ± standard error (SEM).

**Figure 4 pathogens-13-00097-f004:**
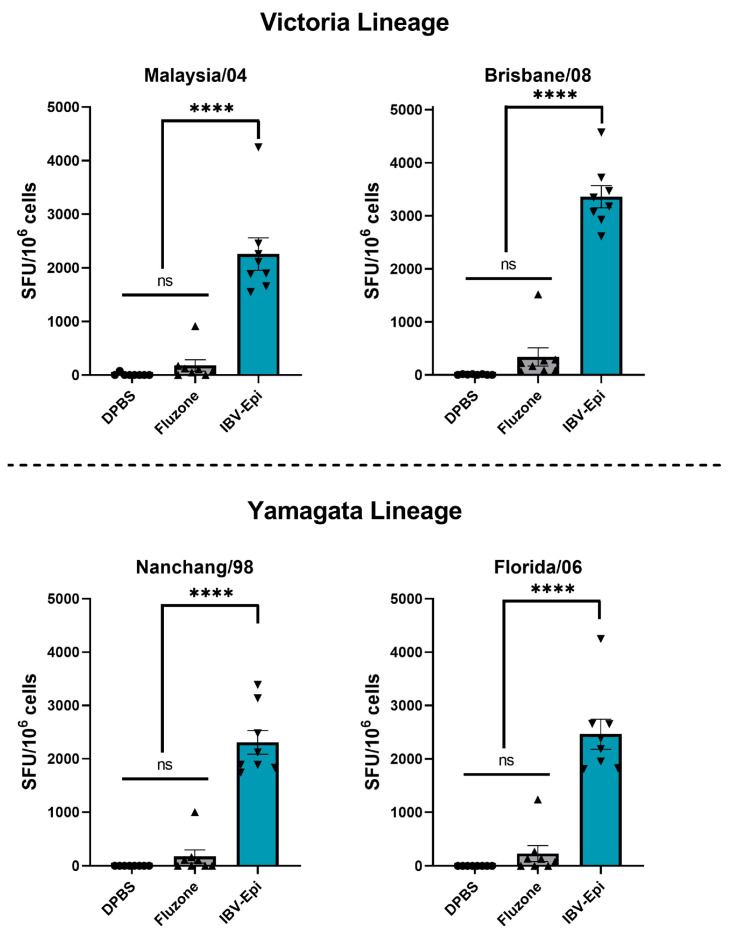
**Cellular responses after vaccination.** Splenocytes were isolated two weeks after boost immunization and screened for antigen-specific T cells by IFN-γ ELISpot. Splenocytes were stimulated with an overlapping peptide array encoding the HA protein of B/Malaysia/2506/04, B/Brisbane/08, B/Nanchang/12/98, or B/Florida/04/06. Data are presented as the mean ± SEM (*n* = 8 mice/group; one-way ANOVA with Tukey’s multiple comparison; ns: not significant, **** *p* < 0.0001).

**Figure 5 pathogens-13-00097-f005:**
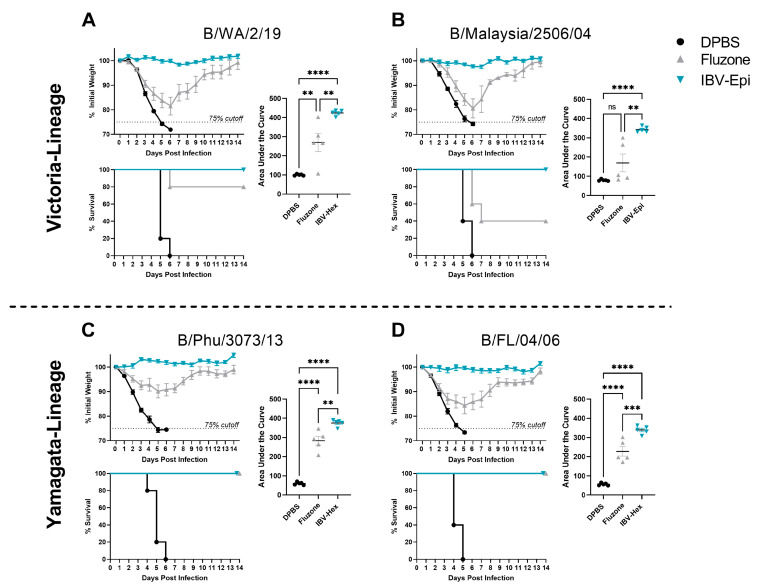
**Protection against lethal challenge with Victoria-Lineage and Yamagata-Lineage IBV.** Mice were immunized twice with IBV-Epi, Fluzone, or DPBS then challenged two weeks after the second immunization with 100MLD_50_ of (**A**) B/WA/2/19 (**B**) B/Malaysia/2506/04 (**C**) B/Phu/3073/13 or (**D**) B/FL/04/06. In each challenge sub-figure, daily weight loss is shown at the top left, the calculated area under the curve of the weight loss plots is shown on the right, and overall percentage of animal survival is indicated on the bottom left. Mice were monitored daily for 14 days and mice that lost ≥ 25% of their initial body weight (dotted line) were humanely euthanized. (*n* = 5; AUC statistical analysis was determined using a one-way ANOVA with Tukey’s multiple comparison follow up test; ** *p* < 0.01, *** *p* < 0.001, **** *p* < 0.0001). Data are presented as the mean ± SEM.

## Data Availability

The Epigraph vaccine designer algorithm used in this study is freely available at https://www.hiv.lanl.gov/content/sequence/EPIGRAPH/Epigraph.html (Accessed on 6 December 2017). All sequences used to create the Epigraph immunogens are freely available through the Influenza Research Database at https://www.fludb.org/brc/home.spg?decorator=influenza (Accessed on 21 January 2024). All other relevant data will be provided by the corresponding author upon request.

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
