# Peer review of "Multivalent Epigraph Hemagglutinin Vaccine Protects against Influenza B Virus in Mice"

_pathogens, 2024, doi:10.3390/pathogens13020097_

Round 1

Reviewer 1 Report

Comments and Suggestions for Authors

Comments for the author of Pathogens manuscript pathogens-2815766:

The author of the Pathogens manuscript “Multivalent Epigraph Hemagglutinin Vaccine Protects Against Influenza B Virus in Mice”, present their recent efforts toward developing vaccines against influenza viruses that have broad reactivity.  Specifically, this manuscript describes the use of computationally designed Epigraph hemagglutinin proteins that target both lineages of influenza B viruses.  They compare this vaccine with the commercial vaccine that includes individual isolates representing the two lineages and demonstrate increased breadth of antibody and T cell immunity with their Epigraph approach.  Importantly, Epigraph-vaccinated mice are protected against lethal challenge with influenza B virus, monitored here using weight loss and survival.  They conclude that the Epigraph approach represents a potential universal vaccine against influenza B virus.  Below are some comments that I would like the authors to address as they revise the manuscript.   

General Comments:

  1.  The manuscript presents an excellent description of the method for producing the vaccine immunogens and a thorough description of the positives and negatives associated with the antigens expressed.
  2. The manuscript fully characterizes the HA proteins expressed and details both expression (Western blot) and functionality (resetting) from the viral vector used.
  3. Use of FluZone as a comparison is a positive for this vaccine study.
  4. Characterization of T cell recognition is another positive for this manuscript.
  5. Can the authors comment on why the T cell responses against FluZone were so low.  Since there are many differences between FluZone and the viral vector-delivered HAs used in this study, it seems the differences in T cell responses are justified, and it might help the reader if the authors could clarify this point.
  6. Have the authors sequenced the mouse-adapted viruses to identify and document any changes in the HA that are associated with this mouse adaptation?  This is not something that takes away from the design or the conclusions drawn, but it is an important piece of information that should be presented.  It might also help explain the differences between Fluzone and Epigraph with the matched strain.
  7. Have the authors evaluated the antibody isotypes induced to determine whether Fluzone and Epigraph differ in the quality of the antibody response.  The differences in weight loss profiles between the two vaccine types could be explained by differences in antibody isotypes, as well as antibody levels.  This is of particular interest because HI and MN antibody levels for B/WZ/2/19 were lower in Epigraph than titers against B/Phu/3073/13 were against FluZone,      yet weight loss was not observed after B/WA/2/19 challenge.  While the authors acknowledge the fact that T cells may make a contribution, it would still be interesting to see if there is a higher level of antibodies of the IgG1 isotype (Th2-like) after Epigraph.  Since the vaccines were all delivered intramuscularly, this minimizes the potential for T cells to be present in the lung and ready to eliminate a large dose of virus (100 MLD50), and leans toward the first wave of protection from weight loss being associated with antibodies.

Author Response

Dear Reviewers,

Thank you for your insightful comments and suggestions. We believe the manuscript is much more complete and significantly improved after revisions. Please see the responses to your comments below. You will find that we corrected the manuscript and included the inserted text in the response in order to help you identify improvements. We thank the reviewers for their time and effort.

Sincerely,

Authors

Reviewer #1:

Comments for the author of Pathogens manuscript pathogens-2815766:

The author of the Pathogens manuscript “Multivalent Epigraph Hemagglutinin Vaccine Protects Against Influenza B Virus in Mice”, present their recent efforts toward developing vaccines against influenza viruses that have broad reactivity.  Specifically, this manuscript describes the use of computationally designed Epigraph hemagglutinin proteins that target both lineages of influenza B viruses.  They compare this vaccine with the commercial vaccine that includes individual isolates representing the two lineages and demonstrate increased breadth of antibody and T cell immunity with their Epigraph approach.  Importantly, Epigraph-vaccinated mice are protected against lethal challenge with influenza B virus, monitored here using weight loss and survival.  They conclude that the Epigraph approach represents a potential universal vaccine against influenza B virus.  Below are some comments that I would like the authors to address as they revise the manuscript.   

General Comments:

  1.  The manuscript presents an excellent description of the method for producing the vaccine immunogens and a thorough description of the positives and negatives associated with the antigens expressed.
  2. The manuscript fully characterizes the HA proteins expressed and details both expression (Western blot) and functionality (resetting) from the viral vector used.
  3. Use of FluZone as a comparison is a positive for this vaccine study.
  4. Characterization of T cell recognition is another positive for this manuscript.
  5. Can the authors comment on why the T cell responses against FluZone were so low.  Since there are many differences between FluZone and the viral vector-delivered HAs used in this study, it seems the differences in T cell responses are justified, and it might help the reader if the authors could clarify this point.

This is an excellent point that we have expanded on in the discussion, which reads as follows, “In addition to strong cross-reactive neutralizing antibody responses, mice vaccinated with IBV-Epi showed robust T cell-mediated immunity against four divergent IBV strains from both the Victoria-like and Yamagata-like lineages while the inactivated split virus comparator vaccine, Fluzone, demonstrated modest induction of T cell-mediated immunity.  The variation of these responses may be due to differences in the vaccine platform. Viral vectors are well known for inducing potent adaptive immunity through intracellular antigen expression and activation of MHC pathways through direct and cross-presentation of the delivered immunogen [47]. In contrast, inactivated virus vaccine platforms do not infect cells to facilitate MHC presentation and priming of CD8+ T cells. This often leads to limited T cell-mediated immunity against influenza after vaccination [48]. Our previous research has assessed T cell activation of the adenoviral vectored Epigraph HA proteins targeting swine H3 IAV compared to an adenoviral vectored vaccine encoding WT swine H3 IAV. While the adenoviral vectored WT HA showed significant T cell activation to the matched strain, immunization with the Epigraph HA proteins demonstrated an increased breath and magnitude of T cell responses against both swine H3 and human H3 IAV [28]. This suggests that the adenoviral vector is contributing to T cell activation, but the breadth of T cell responses against divergent strains is likely due to the Epigraph design. Additional studies directly comparing the Epigraph HA proteins to WT IBV HA proteins are necessary to determine if this same effect is observed with IBV.” (lines 507-525).

  1. Have the authors sequenced the mouse-adapted viruses to identify and document any changes in the HA that are associated with this mouse adaptation?  This is not something that takes away from the design or the conclusions drawn, but it is an important piece of information that should be presented.  It might also help explain the differences between Fluzone and Epigraph with the matched strain.

Indeed, we have previously sequenced two of the four mouse-adapted viruses (B/Phuket/3073/2013 and B/Florida/04/2006; PMID: 35746770) used in this study and interestingly observed no changes in the HA protein after ten passages of mouse-adaptation. We have included this reference in the materials and methods. Given that there were no observed mutations in the HA protein of these mouse adapted viruses, we deemed it unlikely that the protection against B/Phuket/3073/2013 in the IBV-Epi immunization group was only mediated by antibody responses, and induced cross-reactive T cell responses likely played a significant role to the enhanced protection observed. It is worth noting that, though mice immunized with Fluzone demonstrated more significant weight loss than the IBV-Epi group, these mice were also partially protected compared to unimmunized mice. We have taken special care to highlight this point more clearly in the discussion to avoid over conclusion/oversight of these results. The reworded section reads as follows, “…mice immunized with Fluzone demonstrated significant weight loss and mortality after Victoria-like lineage challenges and partial protection against Yamagata-like lineage challenges compared to the unimmunized control mice.” (lines 530-532).

  1. Have the authors evaluated the antibody isotypes induced to determine whether Fluzone and Epigraph differ in the quality of the antibody response.  The differences in weight loss profiles between the two vaccine types could be explained by differences in antibody isotypes, as well as antibody levels.  This is of particular interest because HI and MN antibody levels for B/WZ/2/19 were lower in Epigraph than titers against B/Phu/3073/13 were against FluZone,      yet weight loss was not observed after B/WA/2/19 challenge.  While the authors acknowledge the fact that T cells may make a contribution, it would still be interesting to see if there is a higher level of antibodies of the IgG1 isotype (Th2-like) after Epigraph.  Since the vaccines were all delivered intramuscularly, this minimizes the potential for T cells to be present in the lung and ready to eliminate a large dose of virus (100 MLD50), and leans toward the first wave of protection from weight loss being associated with antibodies.

Unfortunately, we did not assess this in our current study, but we agree that this is a very insightful point that could provide a better understanding of the protection observed. While we did not directly isotype antibodies elicited after vaccination, several studies using recombinant adenoviral vectored vaccines indicate higher induction of IgG2a (Th1-like; classic antiviral T helper cell) compared to IgG1 isotype (Th2-like) (PMID: 36439179 PMID: 34290317 PMID: 34952759). Based on these previous results, we hypothesize that our vaccine is eliciting a similar bias in responses. While this is outside of the scope of the current study, we are excited to pursue this in additional studies where we build off the findings observed in this study to characterize the contribution of the individual Epigraph immunogens and compare these responses to WT HA delivered in an adenoviral vector (and, of course, the commercial comparator vaccine). We hope that future analysis and direct comparison between the listed vaccine platforms will help to delineate possible differences in immune induction after vaccination.

Reviewer 2 Report

Comments and Suggestions for Authors

The Multivalent Epigraph Hemagglutinin Vaccine Protects Against Influenza B Virus in Mice manuscript describes a computationally designed multivalent vaccine featuring Epigraph hemagglutinin proteins that target both the B/Yamagata-like and B/Victoria-like lineages of influenza B viruses in mice. This study compares the immunogenicity and protective efficacy of the designed vaccine with the commercially available Fluzone® vaccine. The results demonstrate the superior performance of the computationally designed Epigraph HA adenovirus type 5 vector vaccine over Fluzone®. The manuscript is well-written, presenting an innovative technology for designing a broadly protective influenza vaccine.

Minor comments

The amino acid sequences of the designed epigraph HA proteins should be accessible in the supplementary materials.

The Adenovirus-based vaccine dosage should be detailed in the Materials and Methods section, specifying virus particle count (VP) and plaque-forming units (PFU). VP count encompasses infectious and noninfectious virus particles, while PFU exclusively represents the value of infectious virus particles. For uniformity and reproducibility, it is imperative to report both counts.

If applicable, the histopathology of lung tissue post-challenge can be presented to evaluate the vaccine's efficacy in safeguarding against tissue damage. Additionally, conducting lung virus titer assessments is recommended, as this provides a comprehensive overview of vaccine protection compared to control groups and the morbidity and mortality evaluation.

Adenovirus vector-based vaccines exhibit a robust cell-mediated immune response, as evidenced by the results compared to Fluzone®. The variation in vaccine platforms in this study may play a role in the broader protection observed with the IBV-Epi vaccine in contrast to Fluzone® (an inactivated vaccine), which elicits minimal cell-mediated immune response. This limitation should be acknowledged in the discussion.

Author Response

Dear Reviewers,

Thank you for your insightful comments and suggestions. We believe the manuscript is much more complete and significantly improved after revisions. Please see the responses to your comments below. You will find that we corrected the manuscript and included the inserted text in the response in order to help you identify improvements. We thank the reviewers for their time and effort.

Sincerely,

Authors

Comments and Suggestions for Authors

The Multivalent Epigraph Hemagglutinin Vaccine Protects Against Influenza B Virus in Mice manuscript describes a computationally designed multivalent vaccine featuring Epigraph hemagglutinin proteins that target both the B/Yamagata-like and B/Victoria-like lineages of influenza B viruses in mice. This study compares the immunogenicity and protective efficacy of the designed vaccine with the commercially available Fluzone® vaccine. The results demonstrate the superior performance of the computationally designed Epigraph HA adenovirus type 5 vector vaccine over Fluzone®. The manuscript is well-written, presenting an innovative technology for designing a broadly protective influenza vaccine.

Minor comments

The amino acid sequences of the designed epigraph HA proteins should be accessible in the supplementary materials.

Unfortunately, because we have a patent pending on these Epigraph HA proteins through the University of Nebraska-Lincoln’s NUTech Ventures, we are unable to publish the sequences at this time.

The Adenovirus-based vaccine dosage should be detailed in the Materials and Methods section, specifying virus particle count (VP) and plaque-forming units (PFU). VP count encompasses infectious and noninfectious virus particles, while PFU exclusively represents the value of infectious virus particles. For uniformity and reproducibility, it is imperative to report both counts.

We have included both the VP and infectious particle (IP) amounts used for vaccination in the methods and included an additional VP:IFU ratio table in Supplementary Figure 1 for easy reference to the reader. The addition to the materials and methods reads as follows,” Infectious units were quantified by Adeno-X Rapid Titer Kit according to manufacturer’s instructions (Takara Bio Company) (Supp. Fig. 1).” (lines 153-154) and  “The hexavalent Epigraph vaccine was a cocktail of 1.67 x 109 vp for each respective HA component (B/Victoria/Epigraph 1: 3.01x107 infectious units; B/Victoria/Epigraph 2: 8.68x106 infectious units; B/Victoria/Epigraph 3: 7.92x106 infectious units; B/Yamagata/Epigraph 1: 6.09x106 infectious units; B/Yamagata/Epigraph 2: 4.23x106 infectious units; B/Yamagata/Epigraph 3: 7.69x106 infectious units).” (lines 185-189).

If applicable, the histopathology of lung tissue post-challenge can be presented to evaluate the vaccine's efficacy in safeguarding against tissue damage. Additionally, conducting lung virus titer assessments is recommended, as this provides a comprehensive overview of vaccine protection compared to control groups and the morbidity and mortality evaluation.

Unfortunately given the rapid turnaround time of 10 days required by MDPI Pathogens, we will be unable to perform and report lung virus titer assessment for these challenges, as this will take a minimum of 6 weeks to complete. However, we are currently in the process of determining protection against challenge of the combined and individual Yamagata-like and Victoria-like Epigraph immunogens to better gauge cross-protective responses compared to the Fluzone control. In these studies, we assess morbidity and mortality through weight loss, death, and will collect lungs for viral enumeration through RT-qPCR and TCID50. We hope to have these studies submitted for publication in the very near future.

Adenovirus vector-based vaccines exhibit a robust cell-mediated immune response, as evidenced by the results compared to Fluzone®. The variation in vaccine platforms in this study may play a role in the broader protection observed with the IBV-Epi vaccine in contrast to Fluzone® (an inactivated vaccine), which elicits minimal cell-mediated immune response. This limitation should be acknowledged in the discussion.

This is an excellent point that we have expanded on in the discussion, which reads as follows, “In addition to strong cross-reactive neutralizing antibody responses, mice vaccinated with IBV-Epi showed robust T cell-mediated immunity against four divergent IBV strains from both the Victoria-like and Yamagata-like lineages while the inactivated split virus comparator vaccine, Fluzone, demonstrated modest induction of T cell-mediated immunity.  The variation of these responses may be due to differences in the vaccine platform. Viral vectors are well known for inducing potent adaptive immunity through intracellular antigen expression and activation of MHC pathways through direct and cross-presentation of the delivered immunogen [47]. In contrast, inactivated virus vaccine platforms do not infect cells to facilitate MHC presentation and priming of CD8+ T cells. This often leads to limited T cell-mediated immunity against influenza after vaccination [48]. Our previous research has assessed T cell activation of the adenoviral vectored Epigraph HA proteins targeting swine H3 IAV compared to an adenoviral vectored vaccine encoding WT swine H3 IAV. While the adenoviral vectored WT HA showed significant T cell activation to the matched strain, immunization with the Epigraph HA proteins demonstrated an increased breath and magnitude of T cell responses against both swine H3 and human H3 IAV [28]. This suggests that the adenoviral vector is contributing to T cell activation, but the breadth of T cell responses against divergent strains is likely due to the Epigraph design. Additional studies directly comparing the Epigraph HA proteins to WT IBV HA proteins are necessary to determine if this same effect is observed with IBV.” (lines 507-525).

Reviewer 3 Report

Comments and Suggestions for Authors

The manuscript is well written and the experiment is nicely organized. I just have some quick questions for the author based on interest.

Major:

1. Do you think using fewer Epigraph B/HA constructs in your IBV-Epi vaccine will make a huge difference in the protection performance?

2. Will it be worth seeking what might be the minimum number of B/HA constructs that are needed in your IBV-Epi vaccine to provide decent protection against IBV infection?

I do not think the author needs to do experiments to answer these questions but I think it might be worth to discuss about it?

Author Response

Dear Reviewers,

Thank you for your insightful comments and suggestions. We believe the manuscript is much more complete and significantly improved after revisions. Please see the responses to your comments below. You will find that we corrected the manuscript and included the inserted text in the response in order to help you identify improvements. We thank the reviewers for their time and effort.

Sincerely,

Authors

Reviewer #3:

Comments and Suggestions for Authors

The manuscript is well written and the experiment is nicely organized. I just have some quick questions for the author based on interest.

Major:

  1. Do you think using fewer Epigraph B/HA constructs in your IBV-Epi vaccine will make a huge difference in the protection performance?
  2. Will it be worth seeking what might be the minimum number of B/HA constructs that are needed in your IBV-Epi vaccine to provide decent protection against IBV infection?

I do not think the author needs to do experiments to answer these questions but I think it might be worth to discuss about it?

We thank the reviewer for their insightful questions. Indeed, we are very curious about these points. Our lab is currently in the process of performing additional studies to assess the contribution of the individual Epigraph HA proteins to increase the breadth and magnitude of protection against the IBV lineages. We aim to submit our findings for publication in the near future. Based on our previous study detailing the Epigraph platform against human H3 IAV, we have observed an additive effect of the individual Epigraph HA proteins to increase cross-reactive responses and hypothesize that this is likely occurring with IBV as well. We have further expanded on these questions in the discussion, which reads as follows, “ We have previous interrogated the contribution of individual Epigraph immunogens targeting human H3 IAV, and observed that the first and second Epigraph HA proteins induced the strongest cross-reactive antibody responses, and the third Epigraph HA protein elicited strong cross-reactive T cell responses. These data indicate that each Epigraph HA protein is uniquely contributing to immunity and a trivalent cocktail is likely necessary to provide broadly cross-reactive responses. In light of these findings, additional studies are currently in progress to determine the participation of individual immunogens in protection and identify the compounding effects of using a multivalent vaccine strategy against IBV.” (lines 498-506).

Round 2

Reviewer 3 Report

Comments and Suggestions for Authors

No further questions need to be addressed.